# Impact of Socio-Economic Status on Accessibility of Dog Training Classes

**DOI:** 10.3390/ani9100849

**Published:** 2019-10-22

**Authors:** Lauren Harris, Tamsin Durston, Jake Flatman, Denise Kelly, Michelle Moat, Rahana Mohammed, Tracey Smith, Maria Wickes, Melissa Upjohn, Rachel Casey

**Affiliations:** 1Canine Behaviour and Research Department, Dogs Trust, 17 Wakley Street, London EC1V 7RQ, UK; tamsin.durston@dogstrust.org.uk (T.D.); jake.flatman@dogstrust.org.uk (J.F.); tracey.smith@dogstrust.org.uk (T.S.); maria.wickes@dogstrust.org.uk (M.W.); melissa.upjohn@dogstrust.org.uk (M.U.); rachel.casey@dogstrust.org.uk (R.C.); 2Campaigns Department, Dogs Trust, 17 Wakley Street, London EC1V 7RQ, UK; denise.kelly@dogstrust.org.uk (D.K.); rahana.mohammed@dogstrust.org.uk (R.M.)

**Keywords:** dogs, behaviour, training, socio-economic status, participation

## Abstract

**Simple Summary:**

Behaviour problems are among the most common reasons owners give for relinquishing their dog to a rehoming centre. Dog training and owner education classes can help prevent behaviour problems, but some people may not attend these due to cost and other barriers, particularly people on low incomes. This study compared the engagement of dog owners recruited in areas with high levels of socio-economic deprivation who were offered free face-to-face dog training classes or an online dog training course. The study aimed to find out whether the online or the face-to-face formats were better at reducing barriers to learning about dog behaviour. There were high dropout rates from both types of courses; none of the participants finished the online course, and 43% of people did not reach the end of the face-to-face classes. A course of paid dog training classes with similar content, running in the same geographic area, had a comparatively low dropout rate (24%). Participants who completed the free face-to-face classes had significantly higher household incomes and were less likely to receive means-tested benefits than participants who dropped out. This evidence suggests that low income dog owners may face other barriers to attending dog training classes, aside from, or in addition to, cost. Future research should investigate people’s reasons for not continuing with dog training courses in order to support the development of training and behaviour advice delivery that is accessible to everyone.

**Abstract:**

Behaviour problems are amongst the most common reasons given for relinquishing dogs to rehoming centres. Some behaviour problems may be amenable to being tackled pre-emptively with classes educating owners on basic dog training and understanding behaviour; however, it is recognised that people with low socio-economic status (SES) may face barriers to attending classes such as affordability, variable working hours, and limited access to transport and childcare. The current study piloted free-to-use dog training and owner education classes in areas with high levels of economic deprivation, both in the traditional face-to-face format and online. It was hypothesised that providing an online dog training course may help people overcome practical barriers by allowing them to complete training modules in their own time. High dropout rates were observed in both formats (online: 100%, face-to-face: 43% dropout). A course of paid dog training classes running in the same area saw a comparatively low dropout rate (24%). Participants who completed the face-to-face classes had significantly higher household incomes and were less likely to receive means-tested benefits than participants who dropped out (household income *p* = 0.049; benefits status *p* = 0.017). This evidence suggests that people with low SES may face non-course fee-related barriers to attending dog training classes. Future research should include a qualitative investigation of people’s reasons for not continuing with dog training courses. Study findings can support the development of training and behaviour advice delivery that is accessible to people with varied socio-economic backgrounds.

## 1. Introduction

Behaviour problems are a common reason given by owners for relinquishing their dogs to rehoming centres. A 2010 study found that “problematic behaviours” were the most common reason reported for the relinquishment of dogs to Dogs Trust centres, with 34.2% of all owners giving this reason, and 55.1% of owners who had previously obtained the dog from Dogs Trust [1]. A US-based project (the Regional Shelter Relinquishment Study) found that 40% of owners relinquishing their dog cited at least one behaviour-related reason [2]. There is evidence that both behavioural advice given to puppy owners [3] and owners attending training classes in early years [4] are associated with a reduced risk of undesired behaviours in later life, which may lead to fewer dogs being relinquished. Identifying approaches which encourage owners to access training or appropriate sources of advice is therefore likely to be important in reducing the occurrence of problem behaviours and risk of later relinquishment.

Dogs Trust Dog School is a programme that delivers classes for owners and their dogs across the country. The aim of Dog School is to help owners build strong, positive relationships with their dogs, in order to help prevent the development of behaviour problems in the future (https://www.dogstrustdogschool.org.uk/). During classes, owners are taught how to train their dogs for everyday activities, such as walking on a loose lead, coming back when called and settling down as needed. In addition, a significant part of the classes involves educating owners about their dog’s behavioural needs to prevent the later development of problems such as separation anxiety, fear and aggression. For example, classes include information on the importance of consistent interactions, how to introduce new people and dogs, and the gradual introduction to new situations and being left alone.

Dog School is affordable to many owners at £55 for an induction session and 5 practical classes. However, it is likely that there are those for whom £55 is a substantial sum for a non-essential service. The current study piloted two free-to-use versions of Dog School, collectively called Introduction to Dog School (ITDS) targeted toward people from low SES (socioeconomic status) communities who may not be able to afford standard Dog School. However, even where a service is free, there may still be barriers which prevent people from low SES communities from attending (for example, limited access to transport and childcare, or unpredictable working hours).

As far as the authors are aware, no studies to date have investigated the accessibility of dog training and behaviour classes to people with low SES. The closest analogue for which there is research available is the participation of parents in free behavioural programmes for children. These programmes involve running classes or workshops for parents which teach parenting skills to reduce behavioural issues in children and are often targeted at disadvantaged families. Some preventative programmes for child behaviour problems have observed lower participation rates from families with low SES [5,6]. Families who dropped out of a child behavioural therapy programme have also been reported to be more likely to be experiencing socioeconomic disadvantage than those who completed the programme [7]. 

Considering the above evidence, it was hypothesised that merely providing free Dog School classes may not make dog training classes completely accessible. The ITDS study gave owners the option to enrol in the course in one of two modalities: face-to-face classes or online modules. It was hypothesised that the online modality could help eliminate barriers such as time schedule conflict and access to transport and childcare, as the training could be completed in people’s own time from home. However, there is evidence that online learning courses are prone to experiencing poor engagement and high rates of dropout [8,9]. 

The current study aimed to investigate differences in attendance and changes in attitude toward training and behaviour between an online and a face-to-face dog training and behaviour course. The study also aimed to compare demographic factors relating to SES (household income and benefits status) between participants who completed the face-to-face course and those who dropped out, in order to determine whether socio-economic factors affected course attendance. Attendance at Dog School (DS) classes (the paid training classes run by Dog Trust) taking place in the same geographic region was also measured and compared to ITDS. Finally, changes in attitude were compared between owners attending ITDS classes and standard DS classes.

## 2. Materials and Methods

The study methodology was approved by the Dogs Trust Ethical Review Board (Reference Number: ERB008).

### 2.1. Participant Recruitment

Participants were recruited at Dogs Trust community events in selected target areas with a known high population of people with low SES (Middlesbrough and Hartlepool). Both target areas were in the top 20 English local authority districts, with the highest proportion of their neighbourhoods in the most deprived 10% on the Index of Multiple Deprivation [10]. Community events were held in public spaces such as community centres or town halls. Two members of Dog School staff set up a stand at the chosen community events; attendees of the event either approached the stand spontaneously or were directed towards it by those running the event. Participants were given the choice to sign up either for the online or face-to-face course.

Participants were also recruited from owners attending paid DS classes running in the same geographic area. Two of the Dog School venues were located in Middleborough, and two were located in the nearby towns of Stockton-on-Tees, and Darlington. Classes in Stockton-on-Tees and Darlington were included to increase sample size, and this was justified by the fact that they are included (along with Middleborough and Hartlepool) in the Tees Valley Local Enterprise Partnership (LEP). The Tees Valley LEP was ranked as having the second highest proportion of neighbourhoods in the most deprived 10% in the country on the Index of Multiple Deprivation [10]. All owners involved in the study, both DS and ITDS, had DL or TS postcodes which fell within the Tees valley LEP. Owners attending their first DS class were asked if they were willing to complete pre- and post-course surveys as part of a study to help improve DS services. 

### 2.2. Pre- and Post-Course Surveys

The pre- and post-course surveys both contained identical questions on owners’ attitudes towards training and behaviour. These attitude questions were in the form of a series of 18 statements; for example, ‘In order to have a happy dog, you must show them that you are the “Boss” or “Pack Leader”’. Owners indicted the extent to which they agreed or disagreed with these statements using a 5-point Likert scale. The pre-course survey included two additional questions relating to SES:Does anyone in your household receive/claim means-tested benefits?Is your household income less than £15,400 per year (or £296 per week)?

Means-tested benefits refers to payments by the UK government made to individuals who can demonstrate that their income and capital is below a specific level. The specific income figure used in the second of these questions is the threshold used to classify “low income” by the UK government, it is calculated as 60% of the median household income in the UK (https://fullfact.org/economy/poverty-uk-guide-facts-and-figures/). Both these questions were Yes/No answers, with the option to select “don’t know” or “prefer not to say”. Earlier versions of the survey included a question on educational attainment, but during ethical review the decision was made to remove this question as it was deemed too sensitive. 

The questionnaires were completed online by owners taking the online course (using the survey tool SmartSurvey (smartsurvey.co.uk). The questionnaires were printed and given in paper form to the participants of the face-to-face classes. Responses to the paper questionnaires were then inputted into SmartSurvey by the instructor after the class so that all the responses were stored together. Paper surveys were destroyed after the data were transferred to the online survey tool. Both questionnaires had a cover page with a statement which explained the purpose of the study and informed the owner that participation was completely optional. The statement also detailed how the owner’s data would be stored. Owners could only proceed to the questionnaire once they confirmed that they had read and understood the statement and were happy to continue.

The survey can be obtained by contacting the corresponding author (L.H.). 

### 2.3. Face-to-Face Classes

Face-to-face ITDS classes started the week after the sign-up event, in the same venue. Participants attended a series of three classes (one class a week, at the same time and location each week). The first class was a presentation on the basics of canine behaviour that underpin training; owners did not bring their dogs to this initial class. The following two classes were practical sessions, where owners brought their dogs and practised positive training techniques as demonstrated by an instructor. Participants were given the pre-course survey to complete at the start of the first class, and the post-course survey at the end of the final class. The course content was based on the 6 class paid DS programme but condensed to 3 classes. For a full list of topics covered in the ITDS course, please see “Appendix A”. The DS participants started the course with the same introductory presentation given to ITDS participants, followed by 5 practical sessions. Course content for DS can be found on the Dog School website (https://www.dogstrustdogschool.org.uk/dog-school/). The same topics were covered in DS and ITDS. However, because the DS course was longer, the participants had more opportunity to practice what they had learnt with the guidance of the trainer. Due to the nature of the classes, the course instructor could adapt the content of the course based on the needs of the owner present. So, classes were not necessarily conducted in an identical manner to the course content plan. Both DS and ITDS classes were group sessions with a maximum of 6 dogs per class.

### 2.4. Online Course

Participants were emailed after the sign-up event with a link to the pre-course survey. On completion of the survey, they were sent a link to the online learning platform and login details so that they could sign in to their personal profile on the platform. The online platform contained three training modules which covered the same topics and structure as the ITDS face-to-face classes (see “Appendix A”). Modules were delivered using a mixture of text, interactive quizzes and videos of canine behaviour professionals demonstrating training methods. Online learners had immediate access to all modules and could complete them in any order they chose. Once online learners completed the course, they were sent the post-course survey (N.B. none of the online learners finished the course. Therefore, there were no post-course surveys obtained from online learners). 

### 2.5. Data Analysis

Dropout rates were determined by calculating the percentage of participants who did not reach the end of the course. Chi-square tests were used to investigate differences in household income and benefits status between those who completed the course and those who dropped out. The SES of participants enrolled in ITDS and those enrolled in DS in the same geographic area was also compared in order to determine whether ITDS was reaching its target audience. 

Attitudes towards training and behaviour were measured using the scores from the attitude questions on the pre- and-post course surveys. Reponses were scored on a scale of 0–4: For positive statements (such as, “Dogs learn by being rewarded for offering desirable behaviours”), participants scored 4 if they answered “strongly agree” and 0 if they answered “strongly disagree”. For negative statements (such as, “Dogs sometimes do things that they know are wrong just to annoy us, or out of spite”), the scoring system was reversed. Scores for each of the 18 statements were summed to calculate a total score for each participant (maximum score = 72, minimum score = 0). Wilcoxon rank sum tests were used to determine whether there were significant differences in total scores between ITDS and DS. Wilcoxon signed rank tests were used to determine whether there was significant improvement on total attitude scores in the post-course survey compared to the pre-course survey. A Kruskal–Wallis test was used to compare median ages of dogs enrolled in ITDS (online), ITDS (face-to-face) and DS.

Statistical analyses were performed using R version 3.6.0 (R Core Team (2019), Vienna, Austria). 

## 3. Results

### 3.1. Dropout Rates

Dropout rates were high for both the online and face-to-face ITDS classes: 51 participants started the face-to-face classes, but only 29 completed the course (43% dropout). None of the 32 participants who started the online course completed it (100% dropout). The dropout rates observed in the DS classes running concurrently in the same area were lower; of the 58 DS participants who started classes, 44 completed the course (24% dropout).

The online course had 19 sections, and there was a function which allowed researchers to see how many of these sections each participant had completed. Only 12 owners completed 1 or more of these sections (eight of these completed just one section, the remaining four owners completed two, four, five and seven sections). The fact that none of the participants completed the online course meant that there were no post-course surveys for these participants. Therefore, it was not possible to compare changes in attitude between the two different modalities of ITDS. 

The trainer attempted to contact participants who dropped out of face-to-face ITDS in order to ascertain the reason they could not come to class. Unfortunately, the majority of those contacted did not respond. Of the 13 individuals who did provide a reason for non-attendance, five cited owner illness, four cited work commitments, two stated that the dog was not suitable for classes due to their behaviour (e.g., “stressed by other dogs”), one said that the dog had come into season, and one reported that the dog had been rehomed. 

### 3.2. Differences in Measures of Socioeconomic Status

Participants who enrolled in ITDS were significantly more likely to have lower household incomes, and to receive means-tested benefits than participants who enrolled in DS in the same geographic area (household income: X^2^ = 29.4, df = 2, *p* < 0.0001; benefits status: X^2^ = 18.5, df = 2, *p* < 0.0001; Figure 1B,D, respectively). Of participants enrolled in ITDS, 40% reported annual household incomes of less than £15,400, and 36% said that someone in their household was in receipt of means-tested benefits. Conversely, in DS only 9% of participants reported household incomes lower than £15,400, and 12% said someone in their household was in receipt of means-tested benefits. This suggests that the ITDS programme was reaching the target audience; people with lower SES who may be less able to afford dog training classes. 

Participants who completed ITDS had significantly higher household incomes, and were less likely to receive means-tested benefits than participants who did not complete ITDS (household income: X^2^ = 6.0, df = 2, *p* = 0.049; benefits status: X^2^ = 8.5, df = 2, *p* = 0.017; Figure 1A,C, respectively). Of participants who completed the course, 21% had household incomes lower than £15,400, and 17% said that someone in their household was in receipt of means-tested benefits. Conversely, of those who did not complete the course, 50% of participants reported household incomes lower than £15,400, and 46% said someone in their household was in receipt of means-tested benefits. 

There was no significant difference in SES between ITDS participants who enrolled on the online course and ITDS participants who enrolled in the face-to-face classes (household income: X^2^ = 1.3, df = 2, *p* = 0.073; benefits status: X^2^ = 1.5, df = 2, *p* = 0.068).

### 3.3. Attitude Scores

Participants who completed the ITDS course, and participants who completed DS, scored significantly higher on the attitudes section in the post-course compared to the pre-course surveys, indicating an improvement in attitude scores as a result of the course (ITDS: V = 69.6, *p* = 0.004; DS: V = 127, *p* < 0.0001). See Table 1 for median scores in the pre- and post-course surveys. 

Participants enrolled in DS scored significantly higher in both the pre- and post-course survey than participants enrolled in ITDS (Pre-course: W = 1041, *p* < 0.0001; Post-course: W = 1001.5, *p* < 0.0001). The median change in pre- and post-course attitude scores was four for both ITDS and DS. So, unsurprisingly, there was no significant difference in improvement in attitude score between ITDS and DS (W = 675.5, *p* = 0.676). 

### 3.4. Dog Age

The average age of dogs participating in ITDS (online), ITDS (face-to-face) and DS was 5 years, 2 years, and 5 months (respectively). A Kruskal–Wallis test indicated that the difference in age between these three groups was statistically significant (X^2^ = 92.8, df = 2, *p* < 0.0001).

## 4. Discussion

The primary aim of this study was to investigate the accessibility of dog training classes for owners with low SES, by comparing dropout rates between an online and face-to-face format. Dropout rates were high for both the online and the face-to-face modalities of ITDS; engagement with the online course was particularly poor, with none of the participants reaching the end of the course. This meant that there were insufficient data to achieve the second aim to compare attitude change between the online and face-to-face modalities. However, there were sufficient data to compare attitude change between DS and ITDS.

Dog School classes running in the same geographic area had significantly lower dropout rates than ITDS. Participants of DS were also significantly more likely to have higher household incomes, and less likely to receive means-tested benefits compared to ITDS participants. This suggests that there may be socioeconomic barriers to attending dog training classes which cannot be fully ameliorated by removing course fees. Comparisons between DS and ITDS should be treated with a degree of caution due to the differences in class structure. However, the existence of socioeconomic barriers other than course fees is further supported by the finding that owners who completed the ITDS course had higher incomes and were less likely to be in receipt of benefits compared to those who were unable to complete the course. Similar associations between low SES and low attendance have been reported in studies relating to accessibility of free behavioural programmes for children [5,6,7]. Mendez and colleagues [11] found high dropout rates among parents attending a programme of free workshops demonstrating early learning activities aimed at low income families (The Companion Curriculum (TCC)): Only 40% of families attended two or more meetings out of a possible nine meetings, and only 1.13% of families attended all nine meetings. 

It could be argued that poor engagement with ITDS reflects a lack of need for dog training in the target community. However, the authors believe this is highly unlikely: Dogs Trust campaign staff, who run multiple responsible dog ownership events in our target communities every week, anecdotally report a persistent high demand for training and behaviour advice. However, because campaign staff are not canine behaviour professionals, they are not fully equipped to deliver this advice themselves. Previous studies have demonstrated that low SES of owners is a risk factor for several canine behavioural problems (including destructive behaviours, house soiling and aggression) [12], and increases the likelihood of being hospitalized with a dog bite injury [13]. A report of UK hospital data for animal bites found that “hospital admissions for bites and strikes by dogs are three times as high in the most deprived areas of England as in the least deprived areas” [14]. Furthermore, the UK hospital data report found that the Durham, Darlington and Tees area (which covers the areas where ITDS was introduced) had the second highest rate of hospital admissions for dog bites and strikes in the country. A study conducted in Liverpool, UK, found that the more economically deprived an area, the more dogs were owned [15]. Finally, the current study found that participants enrolling in ITDS scored lower than DS participants on the attitudes section of the pre-course survey. This suggests that there may be a lower awareness of up-to-date canine behaviour knowledge and the importance of positive training methods in the demographic that ITDS is targeting. Therefore, this demographic is likely to benefit from classes which encourage positive training techniques. Together, this evidence suggests that there is likely to be a need for dog training and behavioural support in our target areas, but dog owners with low SES are less likely to access these services due to various barriers.

It is important to understand what barriers may be preventing people from attending dog training classes, other than course fees. Unfortunately, when ITDS staff attempted to follow-up with participants who dropped out, there was a low response rate. Of those who did respond, the most commonly reported reasons were owner sickness and work commitments. A study on barriers experienced by low income parents to accessing The Companion Curriculum also found that work schedule conflict was a prominent reason for parents being unable to attend workshops (51.30% of parents reported this barrier); other barriers reported were transport (13.99%), tiredness (12.44%) and child care needs (11.40%) [16]. It is likely that many people with low incomes have unpredictable working hours; a recent report by the Living Wage Foundation found that over 1 million people in low paid jobs in the UK have “volatile pay and hours”, and an additional 1.3 million people have regular pay but unpredictable hours [17]. Further research into the identification of barriers is planned for the next phase of ITDS.

A potential factor which could have increased engagement with DS compared to ITDS is the fact that DS classes were paid for in advance. An economic theory, known as the “Sunk Cost Fallacy”, suggests that people are more likely to continue a course of action if they have already invested time or money into it [18]. This effect could be investigated in future research by comparing participation in a partially funded or subsidised dog training course to participation in a completely free training course. Another difference between DS and ITDS which may have affected engagement is the methods of recruitment. For ITDS, participants were actively recruited at campaigns events, they were not necessarily seeking to enrol in canine behaviour classes before they came to the event. For DS, on the other hand, participants were not actively recruited, many participants find out about Dog School by searching for training classes or puppy classes online. Therefore, participants in DS were more likely to be people who had actively sought out training classes, which may account for the increased engagement of these individuals. The decision to utilise different methods of recruitment was made because the researchers believed that people with low SES would be less likely to contact DS spontaneously; this belief was confirmed by the finding that people who enrolled in DS were of higher SES than those who enrolled in ITDS. Differences in recruitment may also have accounted for the significant differences in age between dogs attending DS and ITDS. Dogs attending DS were on average far younger than those attending ITDS—it is likely that owners of young dogs seeking puppy classes online came across DS in their search. Conversely, people at community events with an interest in training were directed towards the ITDS recruitment stand, regardless of the dogs age. Differences in age may have impacted on motivation to complete the course; older dogs may have been less trainable than younger dogs [19], which may have discouraged people from continuing the course if they could not see immediate results.

The online course was particularly poorly attended, none of the participants reached the end of the course, and the majority did not complete a single module. While the face to face course allowed participants to ask the trainer questions, and receive feedback on their practical skills, the online course did not have these benefits: It is possible that this difference made the online course harder to engage with. Although there has been no prior research specifically relating to online dog training courses, other types of online courses have been shown to be prone to high dropout rates [8,9]. Some people may prefer face-to-face to online learning; one study reported that 62% of students would not be willing to enrol on an online degree programme [20]. Studies have found greater reported course satisfaction and feeling of engagement in face-to-face courses compared to online courses [21,22]. In one study a greater proportion of virtual classroom students reported that they were likely to stop attending class in favour of other activities than traditional classroom students [23]. An improvement to the methodology of the current study would be to include user testing of the online platform with members of the target audience; this could help identify areas where participants were likely to disengage with the course. Particular attention should be paid to the user experience when first starting the course, as many participants did not complete a single module, suggesting that the course could do more to engage participants initially. There is increasing evidence that gamification (the application of typical components found in games, for example, point scoring) of online courses can increase user engagement [24,25]. There is also evidence that giving learners social networking opportunities can increase engagement with online courses [26,27]. Future development of online dog training courses should, therefore, consider the use of gamification and social networking as possible methods of increasing user engagement. The next phase of ITDS will take into account the findings of the current study in a number of ways. It is recognized that understanding the barriers faced by people with low SES is a complex issue; due to the nature of our data it was only possible to run relatively simple statistical tests, which may not have taken into account all factors involved. In the next phase of ITDS, participants will be interviewed in order to collect in-depth qualitative data on barriers to training. The intention is to interview both participants who are unable to attend and those who are able to attend, in order to gain insights from different perspectives. The next phase will also involve user testing and focus groups on the online course within our target communities, the aim of which will be to identify factors which discourage people from continuing with the course. A future aim is to include a limited number of partially funded places on existing DS courses. This may utilise the effects of the “sunk cost fallacy” but will also be a more cost-effective way of delivering training compared to running completely separate classes.

## 5. Conclusions

There are many examples of animal charities offering subsidised veterinary care, for example, subsidised neutering or free microchipping services ((UK examples include Dogs Trust, Royal Society for the Prevention of Cruelty to Animals (RSPCA) and People’s Dispensary for Sick Animals (PDSA)). However, as far as the authors are aware, there are no widespread programmes which offer help with the costs of obtaining dog training and behaviour support. The collective findings from this research will help increase understanding of the barriers to attending dog training classes and support the development of training and behaviour advice delivery that is accessible to people with varied socio-economic backgrounds.

## Figures and Tables

**Figure 1 animals-09-00849-f001:**
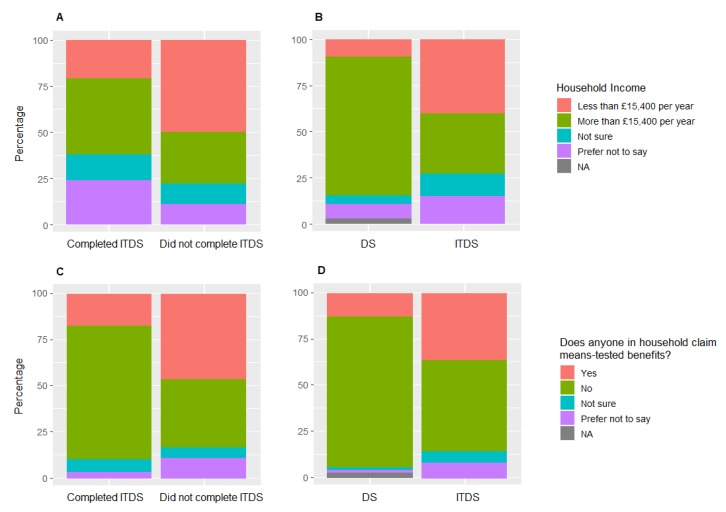
**A**: Comparison of household income between participants who completed Introduction to Dog School (ITDS) and participants who did not compete ITDS; **B**: Comparison of household income between participants who enrolled in DS compared to ITDS; **C**: Comparison of household receipt of means-tested benefits between participants who completed ITDS and participants who did not compete ITDS; **D**: Comparison of household receipt of means-tested benefits between participants who enrolled in DS compared to ITDS.

**Table 1 animals-09-00849-t001:** Median pre- and post-course attitude scores for ITDS and Dog School (DS).

ITDS/DS	Median Pre-Course Survey Score	Median Post-Course Survey Score
ITDS	41	46
DS	47.5	53

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
