# Peer review of "Impact of Socio-Economic Status on Accessibility of Dog Training Classes"

_animals, 2019, doi:10.3390/ani9100849_

Round 1

Reviewer 1 Report

General comments:

The primary aim of this study was to investigate the accessibility of dog training classes for owners with low SES, by comparing drop-out rates between free online classes, free face-to-face classes and payed face-to-face classes. The results are discussed in terms of the socio economic status influencing the higher drop out rate of the free classes as compared to the payed ones. However, from the brief description provided, the courses seem to differ in many more aspects, a part from being or not free and being or not online. These differences make it impossible to attribute the higher drop rate of the free classes to the different socio-economic status of the participants as this may be the effect of the classes being different in their content or execution.

The flows of the study regarding the method involving comparing such different courses should be discussed and the results should be interpreted in the light of these limitations. It cannot be concluded that the different socio-economic status of the participants affected the drop-out rates, since the difference between drop out rate in ITDS and DS may be the effect of the two courses being different in some aspects. This conclusion could only be supported from data collected from participants of one type of course.

Detailed comments:

In the introduction, the authors cite two studies showing that training advices given to owners of young dogs / in early years can help reducing the occurrence of undesired behaviours and, consequently, reduce the risk of relinquishing the dog. It would be important to provide information about the age of the dogs participating in the program. Did the age of the dogs participating in the DS classes differ from the age of the dogs participating in the ITDS classes? May this account for some of the differences in drop out rates (e.g., more owners with adult / older dogs dropped out, maybe because they may perceive that their dogs are less responsive to training?)

Much more information about the 3 different courses (payed dog training classes, free dog training classes and online classes) is needed. It is only written that the program of the free course (ITDS) was based on the program of the payed one (DS) but it is not clear how the program of a 6 week course is condensed in a 3 weeks course (i.e. are some topics not introduced at all? If yes, what? Is less time devoted to each topic? May this affect the learning outcome?). In the supplementary material a quite detailed description of the content of the 3 weeks ITDS face-to-face course is provided, but information about the content of the DS and online course is lacking. This information is essential because any difference among the different classes may potentially explain the differences in drop-out rates (and potentially also in learning outcomes).

From the supplementary material it emerges that the two practical classes offered in the ITDS face-to-face course are group classes with multiple dogs and owners being trained at the same time. Is this also the case for the DS classes? If not, this seems an important difference that may also highly influence the learning outcome and, as a consequence, the drop-out rate.

As for the online course, since it was not a face-to-face (Skype or similar) type of course, it was probably very different from the other two - e.g. it did not include tutored practice with the dogs and feedback on the practice exercises. This, rather than (or in addition to) the fact that it was online per se, may also explain different proportion of drop-out.

Was the improvement in attitude score significant for the participants of the two courses?

It is reported that participants enrolling in ITDS scored lower than DS participants on the attitudes section of the pre-course survey. Apart from suggesting, as the authors write, that there may be a lower awareness of positive training methods and up-to-date dog behaviour knowledge in the demographic that ITDS is targeting, this may also imply that there is less awareness of the usefulness (or need) of training advices in this population. I suggest adding this to the discussion.

Reviewer 2 Report

I found the manuscript very interesting, however I am not surprised by the results.

Some remarks:

You have several abbreviations in the text. Would be helpful to make a list of abbreviations somewhere in the Mat & methods.

Questions:

Was there any socio-economical difference between the online and face-to-face attendants?

Do you have any information about the educational status of the attendants? Could it also affect the results?

L93-94 This sentence belongs to the results/discussion.

L101-113 Participant recruitment. Would be better to mention the number of the recruited participants here. And the drop out rates in the results.

L137- Face-to-face classes. Where were the classes? At the training school? And a question here for the future: What about going and visit the owners & dogs at home?

L264 what do you mean in "....the more DEPRIVED an area...?

Reviewer 3 Report

I appreciated the social relevance and novelty of the paper "Impact of socio-economic status on accessibility of dog training classes". The manuscript is very well written and the study well presented. There are, however, some flaws in the methodology that diminish the strength and validity of the results, and need to be addressed.

The two free-class groups were actively recruited targeting low-income neighborhood, and without taking into adequate consideration their motivation/need to pursue behavior classes. Conversely, the paying group was "passively" recruited selecting people that requested classes, with a probable higher motivation to pursue classes for a perceived need. To eliminate this bias, the Author may have offered free classes to people that reached out to them (the current paying pool).

The statistical methodology used does not allow to account for all the confounding factors that may have influenced the outcomes measured. Using a multivariate model instead of chi-square and Wilcoxon may have allowed to better detect the effect and interaction of the multiple variables like geographical origin of the participants; their income, gender, age; completion of the course; motivation expressed as presence of an undesired behavior... the group studied are very different among them.

Please, find below more detailed comments about the manuscript:

lines 93 – 97: this paragraph about the dropout rate may be moved to Results lines 106 – 110 and 196 – 199: does the “same geographic area” mentioned in line 198 correspond to the “top 20 English local authority districts with the highest proportion of their neighborhoods in the most deprived 10%” mentioned in line 107? This geographical match would help supporting the comparison between free classes and paid-for classes (even though the above mentioned differences in perceived need for classes and active vs passive recruitment should still be addressed and/or discussed); line 121 and elsewhere: the first time “means tested benefits” are mentioned, they should be defined to help the international audience understanding; lines 144 – 145: the different length and density of information between the free and paid-for courses is another factor that introduces variability, which has not been considered in the statistical analysis nor discussed. This factor is even more relevant for the online course. The tendency in online teaching is actually to reduce the amount of information delivered in each class, as well as its length, to keep an optimal attention span; lines 205 – 208: this is the most irrelevant result of the paper, yet relatively modest; lines 239 – 240: as mentioned above, it may be irrelevant that all the dog classes were conducted in the same geographic area if this didn’t correspond to the participants being recruited from the same impoverished districts; lines 240 – 243: I do not think that there are enough elements to state that the findings of this paper “support the hypothesis that there are socioeconomic barriers to attending dog training classes which cannot be fully ameliorated by removing courses fees”. Due to the enrollment methodology used, is not possible to exclude that the two groups had very different motivation and perceived need to enroll in classes. Therefore, in the previous statement I would substitute “support” with “may suggest”; lines 251 – 264: the fact that pets belonging to owners from impoverished areas tend to have more behavior problems, and that they might benefit from training, does not necessarily translate into these dog owners perceiving a need to receive dog training classes. It might even be quite the opposite, and people in financial distress might not perceive dog training classes as one of their priorities or needs; lines 264 – 270: I do agree with the statement in this paragraph. There is certainly a need for dog training support in impoverished area, and there are barriers other than money to deliver this help. This is exactly why I think a statistical model that allows accounting for some of these factors (perceived motivation, length and intensity of the course…) may be better suited.

In conclusion, I believe that the study would benefit from a different statistical analysis, ideally. Using a multivariate model and including some of the variables already in the Author’s database (length of the course, information delivered in every session, dropout rate… any proxy for motivation?) may provide valuable information. However, I understand that on-field studies may not always provide ideal conditions and data. If this should be the case, I recommend to adequately addressing all the above-mentioned limitations in the Discussion.

Round 2

Reviewer 1 Report

I find that the manuscript has significantly improved in quality.

The authors now provide more information about the content and modalities of the different courses and, importantly, provide information about the age of the dogs and discuss the effect of age differences as a potential cause of different drop-out rates.

I think that a sentence recommending caution to the reader in interpreting the findings about differences between drop-out rates in DS and ITDS courses as due to socio-economic factors should be added in the discussion, because the two courses were not identical.

Minor comments:

Line 93: eliminate one of the two "."

Line 123: eliminate one of the two "."

Author Response

Dear Reviewer,

Many thanks for reviewing our manuscript, we have responded to each of your points below:

Point 1: I think that a sentence recommending caution to the reader in interpreting the findings about differences between drop-out rates in DS and ITDS courses as due to socio-economic factors should be added in the discussion, because the two courses were not identical.

Response 1: A sentence to this effect has now been added to the discussion (L 281-283)

Point 2: Line 93: eliminate one of the two "."

Response 2: This change has been made (L 93)

Point 3: Line 123: eliminate one of the two "."

Response 3: This change has been made (L 123)